# ALT-TEXT WITH CONTEXT:
# IMPROVING ACCESSIBILITY FOR IMAGES ON TWITTER

**Nikita Srivatsan**
Carnegie Mellon University
nsrivats@cmu.edu

**Sofía Samaniego**
sofia.samaniego.f@gmail.com

**Omar Florez**
LatinX in AI
omarflorez.research@gmail.com

**Taylor Berg-Kirkpatrick**
University of California San Diego
tberg@ucsd.edu

## ABSTRACT

In this work we present an approach for generating alternative text (or alt-text) descriptions for images shared on social media, specifically Twitter. More than just a special case of image captioning, alt-text is both more literally descriptive and context-specific. Also critically, images posted to Twitter are often accompanied by user-written text that despite not necessarily describing the image may provide useful context that if properly leveraged can be informative. We address this task with a multimodal model that conditions on both textual information from the associated social media post as well as visual signal from the image, and demonstrate that the utility of these two information sources stacks. We put forward a new dataset of 371k images paired with alt-text and tweets scraped from Twitter and evaluate on it across a variety of automated metrics as well as human evaluation. We show that our approach of conditioning on both tweet text and visual information significantly outperforms prior work, by more than 2x on BLEU@4.

## 1 INTRODUCTION

An increasingly important aspect of the social media user experience centers around the sharing and discussion of visual content. However, for users who are blind or have low vision (BLV), this type of content is often inaccessible. One solution to this problem is alt-text, an HTML attribute associated with digital images, which is intended to provide a literal natural language description of an image's content, and can be rendered by a screen reader or braille display. Despite being infrequently included, it is incredibly important from an accessibility standpoint, as it allows users to perceive the content of the image and thus have a better online experience. Some social media websites, notably Twitter, have recently given users the option to write their own alt-text for images that they upload (an example is shown in Figure 1). This approach however offloads the responsibility of making the site accessible onto the users, who frequently either for convenience or lack of awareness simply do not choose to do so (Gleason et al., 2019). We find that as many as 98% of images uploaded to Twitter even after the widespread feature rollout do not have alt-text, to say nothing of the quality of those that do. When a screen reader encounters such an image, it will simply say "Image", leaving the user with no meaningful information about what the image is actually of. Even when images on Twitter do have accompanying user-written alt-text, the quality is inconsistent as not all users are well informed regarding best practices.

Note that while there is a long line of existing research on the broader task of captioning images more generally, alt-text generation for social media is a special case of this task, which in turn comes with its own challenges. Well written alt-text is generally more explicitly descriptive than a high level caption and may emphasize specific details in the image based on context (Kreiss et al., 2022a). See Figure 1 for an example. Furthermore, the distribution of image types on Twitter differs substantially from those found in traditional captioning datasets, and may contain digital artwork, promotional graphics, or screenshots containing text. An additional challenge is that Twitter users are not well trained in the practice of writing alt-text, and therefore native "gold" examples can vary

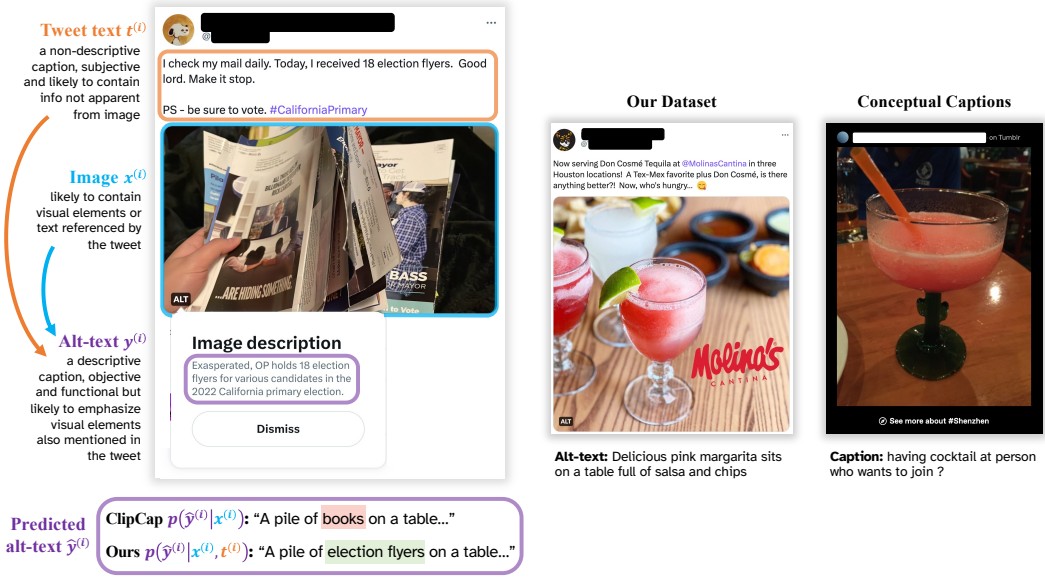

Figure 1: **Left:** An image that requires textual context to write accurate alt-text for. Without conditioning on the tweet text, the election flyers are indistinguishable from books to a traditional captioning system. **Right:** Two similar images from our dataset and Conceptual Captions with their gold labels. The alt-text for the first image is literally descriptive while the second is more colloquial.

substantially in their level of quality. To support continued research in this area, we collect and release a first-of-its-kind dataset of image and alt-text pairs scraped from Twitter.

Our central motivating idea is that despite the challenges associated with this domain, social media images come with additional contextual information that if properly leveraged can significantly improve the quality of automatically generated alt-text. Images posted to social media often require some amount of background knowledge to adequately caption. For example the left image in Figure 1 may easily be mistaken for a stack of books or magazines. However, we do in fact have a source of text that might provide context not present in the visual information contained in the image itself, namely the text associated with the actual post which more specifically identifies the pamphlets as election flyers. We therefore see an opportunity to augment existing captioning systems with textual conditioning in order to allow them to achieve better performance on this challenging task.

One model that is easily modifiable to accept multimodal inputs is ClipCap (Mokady et al., 2021). ClipCap operates using a prefix method, where an image encoder — in this case CLIP (Radford et al., 2021) — produces a vector embedding of the image which is then projected to a sequence of embeddings that occupy the same dimensionality as word embeddings. These (despite not actually being the embedding of any specific language sequence) can then be taken as inputs to a language model decoder — in this case GPT-2 (Radford et al., 2019) — which then autoregressively completes the sequence to produce the caption.

Our approach (shown in Figure 2) supplements the text-like image embedding sequence with an embedding of the tweet text, in the hopes that these two information sources will provide meaningfully non-overlapping signal to the decoder, in order to produce more accurate captions. This reduces the burden on the image encoder to be able to distinguish very similar object types, and allows the decoder access to details that may be very difficult to glean from the visual features.

In summary our contributions are as follows: (1) We collect a novel benchmark dataset for the task of alt-text prediction on social media including both images and surrounding textual context (2) We introduce a novel alt-text generation method that allows for multimodal conditioning with a unimodal language model (3) We perform a quantitative analysis of this approach on our collected dataset, outperforming prior work on captioning (ClipCap, Mokady et al. (2021)) and vision-and-language pretraining (BLIP-2, Li et al. (2023)) by more than 2x on BLEU@4. We also conduct human evaluation which substantiates these findings.

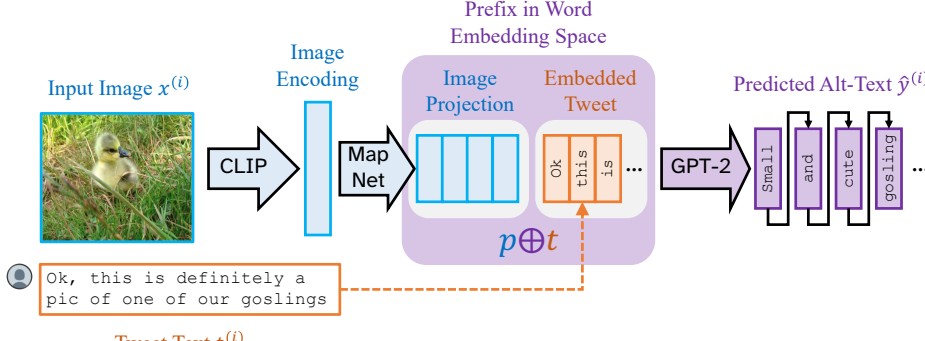

Figure 2: Overview of the alt-text model. An image is encoded via CLIP to obtain an embedding of visual features. This gets projected via a mapping network into word embedding space, where it is then concatenated with an embedded representation of the text from the corresponding tweet. This prefix is passed to a finetuned GPT-2 which autoregressively generates the alt-text caption.

## 2 RELATED WORK

Image captioning in a generic sense is a relatively well studied task. Since the introduction of the COCO benchmark (Lin et al., 2014) a variety of systems have been released, and state-of-the-art quantitative performance has risen across several metrics (Vinyals et al., 2015; Fang et al., 2015; Bernardi et al., 2016). Alt-text generation is relatively understudied by comparison, and there are few established datasets for it.

Kreiss et al. (2022b) collected one such dataset of images with descriptive alt-text and contextual captions from Wikipedia. This corpus was smaller than the one we put forward here at 97k images, and also represents a fundamentally different domain as Wikipedia contains different image types than social media, more frequently has gold alt-text, and has much more carefully curated text captions. They do however demonstrate that conditioning on textual context can improve the quality of generated alt-text, suggesting the value of a similar approach for our setting.

Examining alt-text in the context of social media posts has been studied as well, albeit to an even more limited extent. Twitter A11y (Gleason et al., 2020) is a browser extension based on a pipeline system that primarily works through scraping matching alt-text from alternate sources, only generating it abstractively through a separate caption generator if other methods fail.

Facebook developed and published a system they refer to as Automatic Alt-Text (AAT) (Wu et al., 2017). The approach described does not generate text abstractively, and instead essentially outputs a list of tags indicating various objects detected within the image. While this is arguably more informative than nothing, it's worth emphasizing that this format does not strictly speaking adhere to alt-text best practices. The system has since been refined to include additional details regarding position, count, and size (Meta, 2021). Facebook's AAT has been critically examined, notably by Hanley et al. (2021) who focused particularly on how it describes personal identity categories. They summarize a variety of limitations BLV users experience with it including a lack of detail, a reliance on tags as opposed to fluid natural language captions, and an inability to reason about the contextual salience of various parts of the image.

Chiarella et al. (2020) outlined steps to increase the accessibility of scientific images posted to Twitter. Alt-Textify (Belle. et al., 2022) proposed a method specifically designed for generating alt-text on SVG-based data visualizations and charts, and Chintalapati et al. (2022) put forward a dataset and analysis of visualizations within HCI publications. Li et al. (2020) introduced a method for generating alt-text specifically for mobile UI elements.

One prominent recent example of a more general-purpose system for aligning text with images is CLIP (Radford et al., 2021). While CLIP is not exactly a captioning system, it can be used to score text/image pairs as a way of reranking or selecting captions from a candidate list. Elements of CLIP's architecture have however found success within the pipelines of broader captioning models. One example is ClipCap (Mokady et al., 2021), which uses a mapping network to project a CLIP

image embedding into word embedding space and then feeds the result as a prefix to a language model which outputs the caption. We inherit this basic architectural framework, but extend it to multimodal prefixes which also contain textual context.

While there are other existing datasets for image captioning, they do differ from the one put forward in this paper in a few key ways. One of the most similar examples is Conceptual Captions (Sharma et al., 2018), a dataset of images with short and descriptive text captions. These captions are produced by processing the alt-text descriptions found within HTML to simplify and improve fluency. However, the distribution of images is fundamentally different from that found on social media, and the captions are not generally descriptive enough for accessibility purposes (as a result their captions average only 10 tokens, whereas ours average 40). An example showing this contrast is provided in Figure 1. Furthermore this dataset does not include surrounding textual context, which we demonstrate to be extremely informative for improved alt-text generation.

More recently the emerging subfield of large language models (LLMs) has expanded into exploring multimodal systems capable of conditioning on images interleaved with text. Systems including Flamingo (Alayrac et al., 2022) and GPT-4 (OpenAI, 2023) have gained popularity, although their closed nature makes them difficult to study. Further, there is strong reason to prefer federated systems that are small and controllable by end users rather than through paid opaque APIs that are subject to change without notice and make scientific reproducibility impossible. Open source alternatives such as OpenFlamingo (Awadalla et al., 2023) and BLIP (Li et al., 2022; 2023) have also emerged. These models are capable of captioning images based on few if any in-context examples, despite not being specifically trained on this task. It should be emphasized that these models are however not only incomparable due to their massive and unequal training sets, but are arguably inefficiently overparameterized in their capacity to model a wide variety of tasks beyond captioning. The system we propose instead is trainable on reasonable compute hardware, does not require esoteric prompting strategies, and as we will demonstrate still outperforms BLIP-2 on alt-text generation. Certain large pretrained language models have also been shown to exhibit disability bias (Herold et al., 2022) which is undesirable in general and especially in this context.

## 3 MODEL

At a high level, our model's primary innovation is being able to condition on not just visual information from the image, but also on textual content from the tweet itself. This is in contrast to prior work which captions images in isolation from the text alongside which they appear. We will now describe the multimodal conditioning setup more formally, and over the rest of this section describe our model's architecture for parameterizing the relevant distributions. Suppose that we are given a dataset $X = \{x^{(1)}, ..., x^{(N)}\}$ of images and corresponding tweet text $T = \{t^{(1)}, ..., t^{(N)}\}^1$ as well as gold alt-text descriptions $Y = \{y^{(1)}, ..., y^{(N)}\}$. Our model defines a conditional distribution $p(y|x, t; \theta)$ over the captions given the tweet and image. Having optimized this distribution with respect to $\theta$ over our dataset at train time, at test time our task is to predict $\hat{y} = \arg\max_y p(y|x, t)$. Note that prior work can be thought of as a special case where the tweet $t$ is not considered.

### 3.1 MULTIMODAL CONDITIONING

Our model's decoder must output text in order to generate the alt-text caption. A natural choice for this is an autoregressive transformer-based language model – we specifically opt for GPT-2 (Radford et al., 2019). Conveniently, such architectures' masked self-attention mechanisms make them inherently well suited to conditioning on text. In order to generate alt-text conditioned on the tweet, we can simply feed the tweet to the language model as a prompt and train the model to autoregressively complete it with the corresponding alt-text. The more challenging step is allowing the language model to condition on visual information from the image as well. However, we would also like to avoid fundamentally modifying the decoder's architecture and still be able to leverage unimodal pretraining. In order to solve this, what we require is a way to project image features into text space (or more specifically word embedding space) so that we can directly incorporate them into the tweet prefix. Fortunately, this is something that prior work has addressed. ClipCap (Mokady et al., 2021) learns a mapping network on top of a pretrained CLIP image encoder that projects the visual vector

---

[1]Note $t^{(i)} = t^{(j)}$ if images $i, j$ are from the same tweet.

encoding into a sequence of embeddings that while not directly corresponding to specific words, are still in word embedding space and therefore acceptable by a pretrained language model.

It's worth emphasizing that the crux of this method relies on "translating" image features into a format which the language model can receive as input as if it were a text prefix. An observable artifact of this is that the mapping network often places its outputs near embeddings corresponding to words that are relevant to the contents of the image (Mokady et al., 2021). The prefix can thus be "de-embedded" back into text as a rough means of inspecting what information about the image it encodes. Crucially, this also suggests that further context not present in the image itself could be provided to the decoder in the form of additional text added to the prefix. In this way, we can effectively create a prefix reflecting multimodal information that nonetheless exists in a unimodal space, and can be input to a pretrained unimodal language model.

See Figure 2 for an overview of our model's pipeline. Drawing inspiration from ClipCap's architecture (see Mokady et al. (2021) for details), we pass an image $x$ through a pretrained CLIP image encoder, and project the output to word embedding space using a mapping network. This is parameterized by a multi-layer perceptron that takes in a 512 dimensional vector CLIP embedding and outputs a $k$ long sequence of 768 dimensional vectors (the same size GPT-2 expects for each token). These embeddings constitute a prefix $p$ which is the correct shape to be passed to GPT-2.

Having obtained $p$, a sequence of token embedding-esque vectors, the procedure for combining it with the tweet text to produce a prefix containing both visual and textual information is fairly straightforward. We simply concatenate that projection with the embedded tweet text $t$ (note that they are both sequences in word embedding space) to obtain the complete prefix $p \oplus t$. We can condition on this, treating it as something analogous to a multimodal prompt to our language model, which then autoregressively outputs the alt-text caption $\hat{y}$. Our decoder can therefore condition both on the tweet text, which is the same modality as its output, and the image itself which is not.

To train the model, we simply teacher force and optimize the cross entropy loss between the predicted logits and the gold reference. We backpropagate into the weights of both the mapping network and GPT-2, but leave CLIP frozen both for speed and performance reasons, following Mokady et al. (2021). Further implementation details are provided in Appendix C.

### 3.2 DECODING

After training, there are important choice points for how we decode captions at test time that can significantly impact performance. While ClipCap originally used greedy decoding, we instead opt for beam search with a beam size of 5. Furthermore, as with many language models we noticed that ours occasionally repeated words or short phrases towards the end of captions. We solve this with the trigram blocking strategy described by Paulus et al. (2017). For fair comparison we use this same decoding method for ClipCap in our experiments.

### 3.3 RERANKING

We also experiment with reranking the candidates from beam search by their similarity to the tweet text, as the highest likelihood caption might not necessarily score best under our nondifferentiable evaluation metrics. While the tweet text is not itself a descriptive caption, it may reference specific things in the image that are especially relevant, and therefore high quality alt-text will likely have a significant degree of ngram overlap with it. While the model does get to condition on the tweet text through the prefix, we may wish to ensure that that information does ultimately appear in the final output. In order to steer our generations in this direction, we choose a new best caption among the top 5 candidates returned by beam search based on their ROUGE-L similarity to the tweet text $t$. We also experimented with reranking based on BLEU and CLIP text embedding similarity but found that neither of these performed as well as ROUGE-L.

### 4 DATASET COLLECTION

One contribution of this paper is the collection and release of a large-scale benchmark dataset for training and evaluating alt-text generation systems. We now describe how we collect a publicly available dataset of tweets containing images and corresponding user-written alt-text descriptions.

These tweets were scraped via the public Twitter APIv2 from a 21 day period in the summer of 2022. We focused on English tweets containing still images with user-written alt-text. One important consideration was the removal of duplicates. We noticed that many accounts — especially bots and promotional accounts — would post the same image multiple times, and others (e.g. those tracking extreme weather events or stock prices) often post extremely similar images with only a few words differing between their corresponding alt-text captions. We addressed this by deduplicating based on textual and visual matches. For more details on filtering see Appendix B.

We also took steps to reduce the incidence of names in our data. While many users caption images using the real names of the people in them, predicting a person's name from an image would implicitly require learning to perform facial recognition, something well beyond the scope of this work. Furthermore, from a user privacy standpoint, we would not want our model to be able to output the names of real people, regardless of whether they actually appear in that image. We used the existing NER system of Mishra et al. (2020); Eskander et al. (2020) to identify named entities in our alt-text, and replaced all instances of the `Person` type with the string "person". While this substitutes no additional visual description of the person, and can lead to grammatically cumbersome transformations, we leave further improvements to future work.

This yielded a dataset of 371,270 images. We split into train (330,449), val (20,561), and test (20,260) sets based on tweet ID (i.e. images from the same tweet are assigned to the same split) to prevent leakage of tweet text. The corresponding alt-text captions were on average 144 characters long, which translates to 40 tokens. The tweet text was similarly 152 characters long on average, or 46 tokens. We also hard crop both the alt-text and tweet text at 150 tokens max length. While the raw data cannot be distributed directly in order to protect users' right to be forgotten, we release the dehydrated tweet IDs and media URLs [2].

In order to quantitatively assess the quality of the user-written alt-text descriptions, we randomly sampled 200 tweets from the validation set, and manually inspected them to determine if they were fluent, and descriptive of the image. We found that of the 68.0% of images that still loaded and passed our preprocessing filters (described above), 94.1% were fluently written, and 86.0% contained some form of literal description of the visual contents. Of those that didn't, the majority simply did not provide enough detail or treated the field as a second caption that provided context for what was being depicted, but not a visual description.

## 5 EXPERIMENTS

Having described our method, and our dataset collection procedure, we now conduct experiments to assess how well existing models perform on this new task, and by how much our multimodal system improves over those approaches. This section gives an overview of our experimental setup.

### 5.1 BASELINES

We examine a few key baselines and ablations in our experiments in addition to our full model. The first of these is a **Nearest Neighbor** system, which returns the verbatim alt-text caption associated with the most visually similar image from the training set. We determine visual similarity based on the dot product of the pretrained CLIP features of the images (the same visual encoding used by our full models). The second baseline always returns the corresponding tweet text as the predicted alt-text, which we refer to as **Copy Tweet Text**. This tells us to what extent the tweet text is already a reasonable alt-text caption without transformation. Following our motivation for conditioning on the tweet text, we would expect that this system would occasionally return something informative albeit likely not well formed and certainly redundant from the perspective of a BLV user. This can also be thought of as a simpler variant of our text only system, described below.

Our first neural baseline is **ClipCap**, described previously, using its publicly available implementation. ClipCap is already pretrained on a large captioning dataset, so we evaluate it both out of the box, and finetuned on our own data. For fair comparison, we also use the same updated decoding strategy for our ClipCap experiments. We also compare to pretrained **BLIP-2** (Li et al., 2023) using their OPT (Zhang et al., 2022) 2.7B checkpoint.

---

[2] `https://github.com/NikitaSrivatsan/AltTextPublicICLR`

| Model | Decoding | BLEU@4 | METEOR | ROUGE-L | CIDEr |
|---|---|---|---|---|---|
| **Naive Baselines** | | | | | |
| Nearest Neighbor | - | 0.563 | 1.827 | 5.074 | 1.671 |
| Copy Tweet Text | - | 0.230 | 2.066 | 4.801 | 0.453 |
| **Neural Baselines** | | | | | |
| BLIP-2 (Frozen) | BS | 0.111 | 1.449 | 6.744 | 1.381 |
| ClipCap (Frozen) | BS (NR) | 0.372 | 1.400 | 6.690 | 0.830 |
| ClipCap (Finetuned) | BS (NR) | 0.681 | 2.685 | 8.620 | 1.557 |
| ClipCap (From Scratch) | BS (NR) | 0.696 | 2.643 | 8.483 | 1.588 |
| **Tweet Text Conditioning** | | | | | |
| Ours (Rand Text) | BS (NR) | 0.546 | 2.547 | 7.906 | 1.171 |
| Ours (Text Only) | BS (NR) | 0.693 | 2.459 | 7.498 | 1.738 |
| Ours (Text + Image) | BS (NR) | **1.826** | **3.067** | **8.652** | **7.661** |
| **Decoding Ablation** | | | | | |
| Ours (Text + Image) | Greedy | 0.587 | 2.247 | 7.166 | 1.450 |
| Ours (Text + Image) | Greedy (NR) | 0.469 | 2.168 | 7.774 | 1.520 |
| Ours (Text + Image) | BS | 0.400 | 2.311 | 6.275 | 1.159 |
| Ours (Text + Image) | BS (NR) | 1.826 | 3.067 | 8.652 | **7.661** |
| ClipCap (From Scratch) | BS (NR) + RR | 0.691 | 2.899 | 8.760 | 1.379 |
| Ours (Text + Image) | BS (NR) + RR | **1.897** | **3.257** | **8.818** | 6.791 |

Table 1: Quantitative test results across various metrics (all numbers x100). Our method and Clip-Cap are decoded using beam search with no repeats. **BS** indicates Beam Search, **NR** indicates no repeats, and **RR** indicates ROUGE-L reranking.

Viewing ClipCap as a variant of our model that only includes image features in the prefix begs the question of whether we could similarly include the tweet text, but not the projected image prefix. We therefore also consider a baseline that includes the tweet text $t$ in the prefix, but not the projected image prefix $p$ (referred to as **Ours (Text Only)**). This is effectively a translation model between the tweet text and alt-text; it can transform the style and filter out irrelevant details, but cannot inject new information, and thus serves as an indicator of the performance achievable based only on surrounding context without observing the image itself. We also consider an ablation of our system that substitutes the retrieved nearest neighbor for a randomly chosen tweet, which we call **Ours (Rand Text)**. This lets us measure how robust our system is to errors in the retrieval process and how well it can generalize to cases where a similar neighbor does not exist in train.

## 5.2 METRICS

Most image captioning work has evaluated using quantitative metrics that roughly measure the string similarity between a model's generated caption and a reference gold caption. Since we have user-written alt-text for each image in our dataset, we simply treat these as our gold references to compare to. Specifically, we measure performance on BLEU@4 (Papineni et al., 2002), ME-TEOR (Denkowski & Lavie, 2014), ROUGE-L (Lin & Och, 2004), and CIDEr (Vedantam et al., 2015). It's worth noting however that because the "gold" alt-text captions on Twitter are written by untrained users, they can be noisy, inconsistent in form and specificity, and occasionally do not even describe the image contents (Gleason et al., 2019). As a result, the absolute scores may seem relatively low compared to numbers typically reported on other datasets, although based on qualitative inspection and human assessments they do nonetheless meaningfully correlate with output quality.

Due to this variability, as well as the broader problems with contextless evaluation for this task (Kreiss et al., 2022a), we also conduct human evaluation using Amazon Mechanical Turk. For this experiment, we subsampled 954 examples from our test set (originally 1000 but some tweets had been deleted over time), and asked a group of sighted human annotators to compare the outputs of our multimodal system with two variants of ClipCap for those images (all systems using beam search with no repeats). For more details see Appendix D.

## 6 RESULTS

We now present results on our dataset, both using automatic metrics comparing model output to user-written reference alt-text, and using human evaluation. We also perform qualitative evaluation to analyze our output more subjectively.

| Model | Fluency | Descriptiveness | | Model | Fluency | Descriptiveness |
|---|---|---|---|---|---|---|
| ClipCap (Frozen) | 29.8 | 26.2 | | ClipCap (From Scratch) | 34.2 | 37.7 |
| Ours (Text + Image) | **61.4** | **66.9** | | Ours (Text + Image) | **37.4** | **44.4** |
| Equal Quality | 8.8 | 6.9 | | Equal Quality | 28.5 | 17.9 |

Table 2: **Left:** Results from human evaluation, showing annotator preference for our model vs. frozen ClipCap in terms of fluency and descriptiveness. **Right:** Results showing annotator preference for our model vs. finetuned ClipCap in terms of fluency and descriptiveness.

## 6.1 AUTOMATIC METRICS

In Table 1 we show reconstruction performance on our test set across a variety of metrics. Copying tweet text achieves the lowest performance on all metrics except for METEOR, which is perhaps expected given that tweets are generally not descriptions of the images they accompany. While users will occasionally recycle their tweets into the alt-text field, such examples are filtered out by our preprocessing. Nearest neighbor achieves slightly better performance. This makes sense as there are certain styles of image, for instance selfies, that may appear frequently and have relatively similar captions, although this approach likely makes frequent mistakes when it comes to finer details.

Of the neural methods, we see that frozen ClipCap scores lowest, almost certainly due to domain mismatch in both the distribution of images and text. Finetuning on our dataset does lead to some improvement, bringing ClipCap ahead of the naive baselines on most metrics. Perhaps supporting the hypothesis that our Twitter dataset is fundamentally mismatched from Conceptual Captions, we find that a ClipCap trained from scratch on our dataset performs about on par with the finetuned one depending on the metric. We see similar performance to frozen ClipCap from BLIP-2, which despite its extensive pretraining on data not available to other systems, fails to produce alt-text with sufficient detail and accuracy for this dataset.

However we see by far the strongest results from our approach which combines the mapped image prefix with the text of the tweet as input to a decoder, achieving more than 2x on BLEU and 4x on CIDEr. This is also ahead of the ablation which disregards the image entirely and only uses the tweet text in the prefix, demonstrating that these two sources of information *stack* with one another. The competitive performance of the text-only baseline demonstrates the strong amount of necessary context present in the original post, while the fact that it outperforms merely copying the tweet text emphasizes that alt-text is nonetheless functionally different in many cases from just another caption. Randomizing the retrieved tweet does hurt our model's performance as well. We find qualitatively that under this setup our model is sometimes still able to produce a reasonable caption presumably based solely on the signal from the image features, although it will occasionally insert spurious details from the random tweet, so it's not completely robust to irrelevant input tweet captions.

## 6.2 DECODING ABLATION

Table 1 also shows an ablation of various decoding strategies for our multimodal model. Greedy decoding does poorly, and trigram blocking helps on some metrics but hurts on others. Beam search does not do better than greedy if allowed to repeat trigrams, but the specific combination of beam search without repeats does extremely well. Reranking the top 5 candidates from beam search with ROUGE-L similarity to the tweet text offers minor improvements on most metrics, but hurts performance on CIDEr. We also observe similar rankings of these methods for ClipCap.

## 6.3 HUMAN EVALUATION

Note that while the automatic metrics are informative, the absolute scores are below what these same systems achieve on datasets such as Conceptual Captions (Sharma et al., 2018). We suspect this is largely due to the noisiness of gold references. It's worth emphasizing that Twitter users are not explicitly trained in how to write alt-text, and as a result the level of quality is extremely variable (Gleason et al., 2019). Furthermore, some users write alt-text that functions more as a "second caption" than as a literal description of the image's contents. This means that for an observably large proportion of our data (although the exact amount is difficult to quantify), the gold reference is either undesirable to match or idiosyncratic enough that measuring a candidate caption's similarity to it is

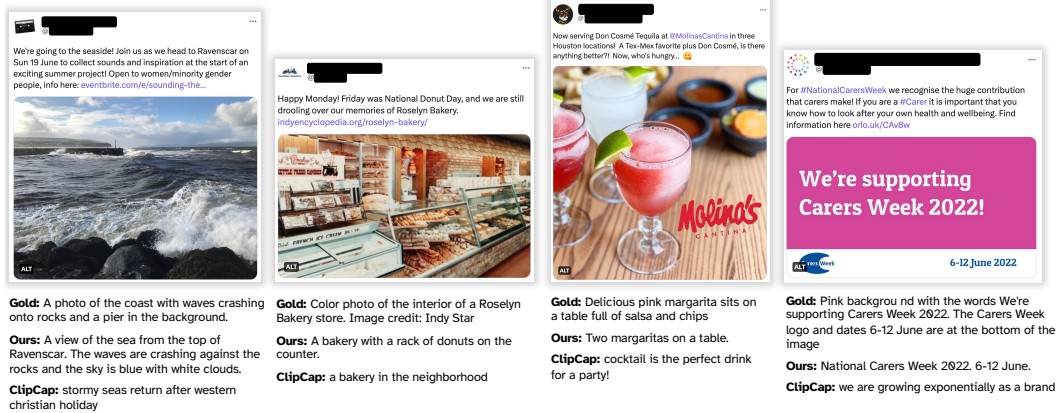

Figure 3: Selected tweets with the user-written alt-text alongside our prediction and ClipCap's. We see that by conditioning on the tweet text, our model is able to focus on relevant details in the images, reference named places, and provide better transcription despite not being trained on OCR.

not truly reflective of how well that caption describes the image. Going through the results manually, we often find instances where the model's output is actually more appropriate as alt-text than what the user wrote, or where they are both functional but phrased differently enough that there is little string overlap, leading to low scores that do not necessarily reflect output quality. We thus conclude that these automated metrics, while informative, are not sufficient for assessing model performance.

We therefore also perform two rounds of human evaluation as described above, with results shown in Table 2. On the left we observe that in roughly two thirds of images the human annotators judged our model's output to be both more fluent and descriptive than that of the frozen ClipCap baseline, and annotators generally agreed on rankings. To the right we see similar rankings albeit with a tighter margin when we compare our system against the ClipCap trained from scratch on our data. This matches the findings of our automated metrics, and supports our hypothesis that conditioning on context from the tweet yields higher quality descriptions.

## 6.4 QUALITATIVE INSPECTION

In order to analyze what specific types of differences lead to these improvements and also what kinds of errors the model still makes, we now describe some observations from qualitative inspection. Figure 3 shows some example tweets from our dataset, accompanied by the actual user-written alt-text as well as our model and a frozen pretrained ClipCap's predictions. We generally see that our system is able to focus on contextually relevant details, such as the donuts in the bakery. It's also able to leverage text in the tweet to more accurately transcribe digital text in graphics. However it does at times still struggle with counting, and sometimes omits thorough visual details. ClipCap provides reasonable albeit simplistic captions for more natural images, but struggles with images that fall farther outside the Conceptual Captions distribution. It is also unable to incorporate information from the tweet itself, or use that to influence which parts of the image to focus on.

## 7 CONCLUSION

In this work we presented an approach for alt-text generation that conditions both on image features and the raw text of the corresponding social media post, and collected a new dataset for this task from Twitter. Despite limitations discussed in the Appendix below, we believe this work is an important step forward and has potential for various downstream applications. While the most obvious use case is to simply use the model to produce alt-text for images that do not have any, we could also see this approach being used in concert with human annotators, perhaps providing a starting point for captions. Additionally, since our model can form a conditional likelihood on the caption, we could use it to flag poorly written alt-text and notify users to reread the guidelines and rephrase their description. In summary, we believe this is an important task and hope that our work encourages future research that leverages the surrounding social media context in which these images appear.

## ETHICS STATEMENT

Given the limitations of our approach (discussed in more detail in Appendix A) we find it critical to emphasize that simply applying this method as-is to producing alt-text captions in an actual social media production setting would be premature and likely harm the user experience for BLV users. Accessibility technology must demonstrably be of more utility than other alternatives in order to justifiably be adopted, and while we hope that our work is a step in that direction, we have not yet conclusively determined that that bar has been passed. The research we perform here has the potential to be co-opted by social media platforms as a cheaper alternative to hiring manual alt-text writers, or having users write their own alt-text as is Twitter's current approach, which would arguably worsen the online experience for BLV users whose accessibility needs are already to a large extent unmet in this regard. Furthermore as with any open-ended text generation system trained on social media, our model does have the capacity to produce toxic language, regurgitate sensitive user information, and produce insensitive and harmful captions for personal photos.

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

# A  LIMITATIONS

This work still has several limitations which are important to acknowledge. The dataset we collected and trained our model on is variable in quality as the gold references are user-written, and this necessarily introduces a gap between the training objective and actual alt-text best practices. It could also be improved with more sophisticated identification and removal of bots, and using entity linking instead of just named entity recognition to provide more finegrained descriptions of people in lieu of their names. The model itself, while it frequently produces reasonable captions, still at times makes mistakes regarding specific details, is imprecise at transcribing text found in images as it does not contain an OCR system, and occasionally outputs the tweet text verbatim which despite at times serving as a reasonably descriptive caption is nonetheless redundant. Since our system conditions on an accompanying social media post, it is not easily adaptable to settings where images appear without any textual context, or where the specific section of text meant to correspond to the image is ambiguous such as in a long form web post or news article. We also rely on having a pretrained image encoder which means that the visual features extracted cannot be informed by the context, they instead must feed into the language model as independent information signals. Finally, it's worth emphasizing that our human evaluation is performed using sighted reviewers, and we therefore cannot make any claims about the downstream efficacy of our method without a larger scale study using BLV users which was unfortunately beyond the scope of this work.

# B  DATASET COLLECTION

## B.1  FILTERING

Reply tweets were included, but retweets were not. We filtered based on tweets containing still images, as alt-text for GIFs tends to be of very low quality and requires different modeling approaches. We also restricted our search to only include tweets which Twitter's language identification model categorized as English, thereby also ensuring that they all contained at least some amount of tweet text. Additional preprocessing included stripping all leading phrases such as "Image of" or "Photo of" (as these are generally considered redundant for alt-text purposes), removing any examples with alt-text that were identical to the text of the tweet itself or simply contained placeholder text such as "Image", and removing any examples with alt-text containing URLs, user handles, and hashtags. We also only included alt-text that was at least 4 space-separated tokens long, as text shorter than this is rarely sufficiently descriptive.

## B.2  DEDUPLICATION

We only retained the oldest of any images that contained identical alt-text. This largely eliminated any exact duplicates posted multiple times by spammy users. Next we identified clusters of visual matches based on their downscaled pixel overlap. To do this, we resized images and center cropped to a max size of $32 \times 32$ pixels, computed all pairwise pixel diffs within our data, and agglomeratively grouped them into clusters using a $< 100$ differing pixels threshold to the nearest match as a cutoff for cluster membership. Within each cluster, we only retained the oldest tweet. It's also worth mentioning that while we did consider further deduplication based on SSIM (Snell et al., 2017) similarity and ngram similarity of the alt-text itself, these approaches proved computationally infeasible.

# C  IMPLEMENTATION DETAILS

We train our models with a batch size of 100 and an initial learning rate of $1e - 4$ using the Adam optimization algorithm (Kingma & Ba, 2015). Our prefixes are of size $k = 10$. This allows training to fit on a single A6000 GPU. Our implementation is written in PyTorch (Paszke et al., 2019) and inherits some code from the ClipCap (Mokady et al., 2021) repository. Experiments train in roughly 6-18 hours depending on the model, the hyperparameter settings, and when early stopping happens (we perform early stopping based on model likelihood on the held-out validation set).

## D    HUMAN EVALUATION SETUP

For each example, turkers were shown the image, the original tweet the image was from (including its text and any other images it contained), and the two candidate alt-text captions. They were never shown the original user-written alt-text. After providing them with a detailed explanation of alt-text and the list of desirable criteria according to Twitter's accessibility guide, we asked two questions: first which candidate alt-text was more fluent, and second which had a better literal description of the image. The models were also anonymized and their order was shuffled.

In order to ensure high quality labels, we performed a qualification round on our reviewers. Considering only turkers who had completed over $10,000$ HITs, we showed them three example images where the two candidate alt-text captions were the gold user-written reference and a manually written, intentionally clearly lower quality caption. Only reviewers who successfully picked the real user-written caption for both fluency and descriptiveness for all three examples were allowed to participate in the final larger scale comparisons between our model and frozen ClipCap, and our model and ClipCap from scratch.

