# OpenReview forum: "Alt-Text with Context: Improving Accessibility for Images on Twitter"
_ICLR.cc/2024/Conference — ICLR 2024 poster_

### Official Review · Reviewer_2NmJ · 2023-10-18

**Soundness:** 2 fair
**Presentation:** 2 fair
**Contribution:** 2 fair
**Rating:** 5
**Confidence:** 4

**Summary:**

The paper presents a method to generate accessibility captions for images shared on Twitter. The proposed approach combines CLIP embeddings for the images, as well as additional context that is included in the tweet’s text to create an embedding that is then fed to GPT-2 to generate the accessibility description. The paper’s evaluation demonstrates that the proposed approach can outperform naive and neural-based approaches like ClipCap and BLIP-2.

**Strengths:**

First of all, I would like to applaud the authors for working on this important and timely problem. I believe that this research is very important and can have the potential to improve the lives and online experience of many people with visual impairments. Overall, I believe that this research focuses on an important problem, and there is potential for a big impact. Second, the paper collects a large-scale dataset of images and user-generated accessibility captions from Twitter; this dataset is far bigger than previous research efforts focusing on similar research problems. Third, I believe that the paper’s approach is a simple, creative, and effective method to combine CLIP embeddings, tweet text, and LLMs to generate accessibility captions for images. The paper’s approach is easy to understand and combines important features for generating contextual and useful accessibility descriptions for images shared on Twitter. Finally, I like that the paper evaluates the performance of the proposed approach both with quantitative metrics as well as via a user study that aims to assess how users perceive and compare accessibility descriptions generated from the proposed method and baseline approaches.

**Weaknesses:**

I have several concerns with the paper, mainly related to the lack of gold standards for accessibility captions, the lack of important and adequate methodological details, the paper’s evaluation, the paper’s approach to releasing data, and the paper’s ethical considerations.

First, there is a disconnection between the paper’s motivation and how the paper evaluates the performance of the proposed method. I agree with the paper’s motivation that the user-generated accessibility captions are of questionable quality, given that most users are unaware of best practices for generating accessibility captions. On the other hand, however, the paper collects user-generated accessibility captions and treats them as gold standards (i.e., ground truth captions). This is problematic as in the evaluation, the paper compares the generated captions from their approach and compares them with captions that are of questionable quality. Therefore, it is not clear what is the actual performance of the proposed methods. A way to potentially mitigate this issue is to apply the proposed approach to other datasets released by previous research that include gold-standard captions (i.e., captions that adhere to the best practices for generating accessibility descriptions for images).

Second, I am puzzled about how the BLEU@4 score is calculated in the evaluation. To the best of my knowledge, the BLEU score ranges from 0 to 1 and aims to assess the precision of the n-grams included in the generated text compared to the ground truth. In the paper’s evaluation, the paper mentions that the proposed approach has a BLEU@4 score of around 1.8. I suggest to the authors to clarify how they calculated the BLEU scores (e.g., if they used a modified version) and better describe how we can interpret these BLEU@4 values.

Third, the paper lacks important and adequate details on the paper’s methodology. Particularly, the paper refers to several appendices so that the reader can get more information, however, there are no appendices in the manuscript. This hurts the readability of the paper and does not allow us to assess the quality and robustness of the presented results in the paper. I suggest including the appendices so that we can understand how the paper conducted various steps of the research. In particular, I would have liked to read more on how the paper conducted the user study, how they recruited users, what is their background and expertise with regards to the best practices for generating accessibility descriptions, etc. All these details are paramount for understanding the quality of the presented research.

Fourth, I have some concerns about the paper’s approach to releasing the dataset. Given the recent changes to Twitter’s API, it became extremely hard to rehydrate tweets based on their IDs. So by simply releasing the Twitter IDs and the media URLs, interested researchers will not be able to reproduce the paper’s results and further use this dataset for further studying this problem. I suggest to the authors to consider releasing more attributes from the dataset (specifically the tweet’s text) so that interested researchers can reproduce the paper’s results without relying on the closed and expensive new Twitter APIs.

Fifth, the paper does not properly explain how the qualitative assessment is done (in Section 6.4), which does not allow the reader to understand if it’s done in a systematic way or how representative/generalizable the insights are. I suggest to the authors to include more details on how the samples for the quantitative analysis are selected and, more importantly, how the qualitative assessment is undertaken (e.g., are the people experts in the domain of accessibility description generation, are they aware of the best practices, etc.)

Finally, the paper does not discuss the ethical considerations when conducting this research. This is important as the paper conducts a user study and shows participants’ images shared on Twitter. For instance, did the paper ensure that there are no harmful images in the dataset and that no participants were exposed to harmful information?

**Questions:**

1. What is the rationale for using user-generated captions as gold standards and do you have an idea how this affects the presented results?
2. How is the BLEU@4 score calculated and did you use a modified version of the metric?
3. How is the user-study conducted and what are the background/expertise of the recruited participants? Also, have you obtained an IRB approval before conducting the user study? How did you ensure that participants were not exposed to harmful information?

**Details Of Ethics Concerns:**

The paper lacks a discussion of their ethical considerations. It's unclear whether the research participants were exposed to harmful images shared on Twitter.

---

> ### Author Response · Authors · 2023-11-16
>
> Thank you for your time and feedback! It’s true as we note that the alt-text found on Twitter is largely written by untrained users. We find that in most cases it is functional, but we did want to at least acknowledge that there is some noise in our data, as with any user-generated corpus. To some extent this is an inherent problem with collecting a large-scale alt-text dataset in the first place. If we attempted to crowdsource the alt-text we would face similar problems, and hiring a small number of experts would severely limit scale. And despite some unavoidable noisiness, our dataset is the first of its kind at this size for this task – there weren’t really alternatives to work on instead. We also hope that to some extent this averages out with scale, i.e. over a large enough dataset the underlying signal will be learnable and measurable.
>
> Furthermore this is why in addition to computing automatic metrics comparing model outputs to the gold references, we also perform human evaluation (see below) where we ask sighted users to compare generated alt-text from various systems. The original user-written captions do not factor into this experiment, but we see similar results to our experiments using automatic metrics.
>
> Regarding BLEU score, we are using a publicly available implementation of all automatic metrics from pycocoevalcap, the same one used by ClipCap and several other papers. As indicated in the Table 1 caption, all numbers are x100 (i.e. BLEU ranges 0-100), which is standard practice for reporting in much of the prior literature.
>
> We tried to follow ICLR’s instructions for supplementary material by uploading it separately from the main submission PDF. The appendix is in its own PDF wrapped in a zip file, which you should see listed under “Supplementary Material”. At least from our end it can be downloaded and viewed from the OpenReview submission page. Please let us and the chairs know if you still cannot access it.
>
> Regarding rehydration, we understand your concerns and wish that there was a more convenient way to release our dataset as we do believe in the importance of our research being replicable and easy to follow up on. However we have to balance this with users’ right to be forgotten and the website’s terms of use. If a user no longer wishes for their image and associated tweet to be either online or in an open training set and deletes them, we have a responsibility to comply with that. We’re in a tough situation here as the API was changed while this paper was in progress, but we believe that this approach has the best tradeoff between facilitating reproducibility and respecting user rights. If you have other suggestions for how we could do this, we’re certainly open to hearing them.
>
> The human evaluation was conducted in a systematic way – there’s some information in 5.2 and more in Appendix E (see above). To summarize, we randomly selected 1k examples from our test set and hired annotators from amazon mechanical turk to assess model outputs. Turkers were given detailed instructions on the task, shown contrasting examples of both well and poorly written alt-text, and asked to read a document with further details on best practices according to Twitter’s accessibility guidelines. Before allowing them to participate in the actual study, we also conducted a dummy round on a small number of tweets with two candidate alt-text descriptions each. Only turkers who correctly selected the best alt-text for all of these were allowed into the next round which we report numbers on in our paper.
>
> We consulted with legal support and arrived at the conclusion that for the purposes of this study, the turkers would not be regarded as human subjects, but rather as annotators. This is because we were not observing or gathering personal information from or about them, but rather hiring them to evaluate our data. Based on this we were under the impression that this assessment would be IRB exempt.
>
> We have an ethics section as well as a limitations section in our appendix (see above). We’re sorry those were hard to find. We do definitely agree that there are important ethical considerations with our research, especially with regard to the potential impact on quality of user experience for BLV people. We did also take steps as described to filter out personal information such as names from our corpus. With regard to annotators viewing harmful information, we are to some extent dependent on Twitter’s ability to enforce its own terms of service, and dynamically populated tweets and images such that anything that had been flagged for content in the several months between initial collection and the final round of human eval would not render. Finally we did also provide both a warning and mandatory acknowledgement from annotators before starting that they might encounter NSFW content, and also indicated this in the title of the task listing.

---

> > ### Comment · Reviewer_2NmJ · 2023-11-22
> >
> > I would like to thank the authors for the clarifications on my comments. Many of my concerns are addressed in the rebuttal. However, my main concern still exists. I believe that it is problematic that the paper assumes that the captions from Twitter users are gold standards and that Twitter users follow the best practices for generating accessibility issues. Also, I am not convinced by the argument that the problem is diminished because of the large scale of the dataset. I expect that most of the users will not follow the standards and guidelines when generating the captions, hence it is likely that a large part of the dataset is of questionable quality. Nevertheless, after the rebuttal, I increased the score in my review.

---

### Official Review · Reviewer_1eBv · 2023-10-29

**Soundness:** 2 fair
**Presentation:** 3 good
**Contribution:** 2 fair
**Rating:** 6
**Confidence:** 3

**Summary:**

This paper proposes a method for generating "alt-text" for social media images. In this paper, "alt-text" is explained as a more detailed description and context-specific than a generic image caption.
The proposed method takes an image and the social media text that accompanies the image and outputs the alt-text of the image. The input image is encoded by CLIP and combined with the encoded text. The combined information is then input to GPT-2 to generate the alt-text.
The author has collected tweets and their corresponding images with alt-text to create a dataset for alt-text generation research. The evaluation shows the results of using this dataset to compare the proposed method with several baseline methods.

**Strengths:**

The author points out that in social media, images are often posted in addition to textual information, but there is little information describing the images, and in such situations, information about the images is not conveyed by text-to-speech software for the visually impaired, for example. I agree with this point and understand its importance as a study.

As for the proposed method, its basic structure consists of encoding images using CLIP and generating text using GPT-2. This structure itself is not unique, as it is a concept that has been used in existing research. I believe that the originality lies in the extended part of the basic configuration, where not only the image but also the text to which the image is attached is input.

The fact that an original data set was constructed for this study is commendable. It is also commendable that in the evaluation using this data set, comparisons with various methods were made to show the characteristics of this task.

**Weaknesses:**

As mentioned in the "Strengths" section, the focus on "alt-text" is highly evaluated, but there is room for improvement in that "alt-text" is not clearly defined in the paper.

In the evaluation dataset, the "alt-text" entered by twitter users is used as the correct answer, but it is written as "alt-text captions on Twitter are written by untrained users, they can be noisy, inconsistent in form and specificity, and occasionally do not even describe the image contents" in the paper, and it seems that the authors are discussing the generation method without knowing what "alt-text" is.

Although the difficulty is understandable, I think that the discussion should start with clarifying what "alt-text" is, and then discussing the generation method.

**Questions:**

I wondered if the "alt-text" would be written differently for different people, even for the same image. Is it possible for the proposed method to learn well in such a case?

In addition, although the paper uses crowdsourcing for evaluation, I think it is possible that the evaluation by the viewer of the "akt-text" may also differ from person to person. How did the evaluation go this time?

---

> ### Author Response · Authors · 2023-11-16
>
> Thank you for your time and feedback! We apologize that the definition of alt-text wasn’t made more clear in the paper. Alt-text does in fact have a fairly specific meaning, it’s a well established accessibility feature with generally agreed upon best practices (https://help.twitter.com/en/using-x/write-image-descriptions). Alt-text is meant to be a literal image description that unlike a traditional caption is both more objective and focuses on contextually salient details which may vary with the context in which the image appears (hence our motivation for conditioning on surrounding tweet text). To answer your question, alt-text should ideally not be written very differently from person to person as it is meant to be an objective description that can stand in for the image itself.
>
> It’s true as we note that the alt-text found on Twitter is largely written by untrained users. We find that in most cases it is functional, but we did want to at least acknowledge that there is some noise in our data, as with any user-generated corpus. To some extent this is an inherent problem with collecting a large-scale alt-text dataset in the first place. If we attempted to crowdsource the alt-text we would face similar problems, and hiring a small number of experts would severely limit scale. And despite some unavoidable noisiness, our dataset is the first of its kind at this size for this task– there weren’t really alternatives to work on instead. We also hope that to some extent this averages out with scale, i.e. over a large enough dataset the underlying signal will be learnable.
>
> Furthermore this is why in addition to computing automatic metrics comparing model outputs to the gold references, we also perform human evaluation where we ask sighted users to compare generated alt-text from various systems. The original user-written captions do not factor into this experiment, but we see similar results to our experiments using automatic metrics. To answer your question, we also performed fairly thorough worker vetting for our human evaluation as described in our Appendix, in order to ensure consistency in judgements. We could potentially perform a followup study to more directly measure the extent of inter annotator agreement for the next revision.

---

> > ### Comment · Reviewer_1eBv · 2023-12-04
> >
> > Thank you for taking the time and effort to answer the above questions. This has addressed most of the concerns I felt, so I have raised my score to a 6.

---

### Official Review · Reviewer_xHBk · 2023-10-29

**Soundness:** 3 good
**Presentation:** 3 good
**Contribution:** 3 good
**Rating:** 8
**Confidence:** 3

**Summary:**

This paper studies on an important problem: how to generate Alt-text to help improve the accessibility of social media images. More specifically, the authors collected and cleaned a large dataset of images incorporated with the alt-text labeled by users. With this large dataset, they are able to evaluate the proposed method and some baselines. Overall the proposed solution is not novel for the Machine Learning Community. But the research problem is interesting and meaningful. Also, the collected dataset is important for the community of HCI and Social Media Analysis.

**Strengths:**

1. The collected dataset is important and useful. The data preprocess ensure its usability.

2. The research problem raised in this paper is important.

**Weaknesses:**

1. The novelty of the proposed method is really limited excpet the tweet-text-based reranking.

2. The experiment is somewhat not extensive. For example, from Table 1, it seems that the tweet-based reranking is the most important component. But the authors did not tried to incorporate the reranking with the baselines, which is not fair.

**Questions:**

1. Will the tweet-based reranking improve the baselines like BLIP-2 and ClipCap?

2. Is there some other reranking strategies that you tried? Such as comparing the Clip-based similarity in the representation space?

---

> ### Author Response · Authors · 2023-11-16
>
> Thank you for your time and feedback! We’d argue that the tweet text reranking is far from the most novel thing in our model (as well as not being the primary source of our quantitative gains), rather the way we approach alt-text generation (which is in general understudied) by way of *contextual text conditional* image captioning is in our opinion more new and interesting, and also accounts for more of our numerical performance improvement over prior work. While work exists that has projected image features into text space, there’s much less precedent for our way of creating a prefix from multiple modalities, which we empirically demonstrate is what’s responsible for our gains.
>
> On the topic of reranking though, we’d like to clarify our experiments – and critically emphasize that as stated in our paper, we *did* try reranking on ClipCap and found that it yielded similar results. We’ll try to make that more clear in the next revision. Specifically we found that it improved performance there as well by a similar amount, for example boosting BLEU from 0.681 to 0.793 (we’ll include those numbers in the next revision as well). You’ll notice by comparing the two bolded rows in the table that improvement from reranking itself is similarly small but noticeable for our model also. But we want to strongly emphasize that outside of the ablation at the bottom section of the table, the comparisons between our model and the baselines are apples-to-apples as far as the decoding strategy and therefore fair to directly compare. Column 2 indicates the decoding strategy used for each row, and you can see rows corresponding to our model and ClipCap decoded with the exact same strategy (i.e. no reranking for either). In fact the primary gain comes not from reranking but from the specific combination of beam search with no repeats *which is exactly what we do for ClipCap in all experiments* even though they themselves in their original paper do not.
>
> We did also try various other reranking strategies, including reranking with BLEU instead of ROGUE, reranking with an inverse model that predicts the tweet text from the alt-text, and reranking by the CLIP text embedding similarity as you suggested. These all do worse than ROGUE, but better than not reranking at all. The CLIP text embedding similarity reranker for example gets 1.834 on BLEU. We can add those numbers to the next revision as well.

---

### Meta-Review · Area_Chair_3gX7 · 2023-12-06

**Metareview:**

This paper addresses the task of generating alternative text (alt-text) descriptions of images to improve accessibility of social media/Twitter image-based posts. The contributions are twofold: the authors collect a new and large-scale Twitter dataset for the task, and develop a multimodal model that conditions on both textual information from the post as well as visual signal from the image. While the technical novelty is limited as such multimodal approaches have been investigated before, the dataset is novel (particularly in terms of size) and addresses an important problem.
While the reviewers have raised a number of concerns, these seem to have been largely addressed during the rebuttal, also indicated by the reviewers’ updated scores. However, there are two issues that remain: first, the new Twitter/X API makes it quite difficult to rehydrate tweets, and so it will be difficult for researchers to actually utilise this dataset and/or reproduce this work. I appreciate, however, that the API changed while this work was in progress, and we cannot possibly punish the authors for this. The second and more important issue relates to the quality of the collected dataset. I agree with reviewer 2NmJ that captions collected by untrained users who do not necessarily follow best practices with respect to accessibility are not an ideal gold standard. The authors argue that noise is diminished because of the large scale of the dataset. Existing experiments indirectly attest to the quality of the data; however, more explicit comparisons would have been useful too. For example, reviewer 2NmJ  suggests the authors evaluate their approach against  other datasets released by previous research that include gold-standard captions (i.e., captions that adhere to best practices for generating accessibility descriptions for images). There is a general comment from the authors that there are no existing datasets that are usable for evaluation but it is not clear why, and the authors do not directly respond to the reviewer’s suggestion. Furthermore, the authors mention that “We could potentially perform a followup study to more directly measure the extent of inter annotator agreement for the next revision”. It would have been nice to provide such an analysis as part of the rebuttal so we can immediately assess this. In general, however, there is value to this work and could benefit the community. I would advise the authors to incorporate all of the reviewers’ suggestions into their paper, and add additional experiments that can help one appreciate the significance of this work.

**Justification For Why Not Higher Score:**

There are concerns with respect to the quality of the collected dataset that are not fully addressed with the existing experiments.

**Justification For Why Not Lower Score:**

I think there is still value in this work and it could benefit the community.

---

### Decision · Program_Chairs · 2024-01-16

Accept (poster)